**Data Availability Statement:** Our original data files are available from the figshare repository (doi.org/10.6084/m9.figshare.13536605.v1).

# Learning strategies and their correlation with academic success in biology and physiology examinations during the preclinical years of medical school

**Annemarie Hogh, Brigitte Müller-Hilke**  *

Core Facility for Cell Sorting and Cell Analysis, University Medical Center Rostock, Rostock, Germany

* brigitte.mueller-hilke@med.uni-rostock.de

## Abstract

### Background

Efficient learning is essential for successful completion of the medical degree and students use a variety of strategies to cope with university requirements. However, strategies that lead to academic success have hardly been explored. We therefore evaluated the individual learning approaches used by a cohort of medical students in their first and second preclinical years and analyzed possible correlations with examination scores.

### Methods

107 students participated in our longitudinal survey on cognitive, meta-cognitive and resource-oriented learning strategies using the LIST-questionnaire (Lernstrategien im Studium). The students were surveyed twice while in their first and second year of medical school, respectively and academic performances were assessed as scores obtained in two examinations written shortly after the LIST surveys. Statistical evaluations included comparisons and cluster analyses.

### Results

We here identified four different patterns of learning strategy combinations, describing the relaxed, diligent, hard-working, and sociable learners. About half of the students stayed true to their initially registered pattern of learning strategy combinations while 53 students underwent a change between the first and second surveys. Changes were predominantly made between the *relaxed* and the *sociable* and between the *diligent* and the *hard-working* learners, respectively. Examination results suggested that the *diligent* and *hard-working* learners were academically more successful than the *relaxed* and *sociable* ones.

### Conclusion

Early habits of sociable learning were quickly abandoned however, not in favor of more successful patterns. It is therefore essential to develop interventions on learning skills that have a lasting impact on the pattern of the students´ learning strategy combinations.

**Funding:** The author(s) received no specific funding for this work.

**Competing interests:** The authors have declared that no competing interests exist.

# Background

Students in higher education are expected to autonomously learn, rehearse, and deepen the teaching content conveyed in lectures and seminars. Likewise, they should be able to independently prepare for oral and written examinations which is often described as self-regulated learning [1, 2]. The concept of self-regulation provides a basic principle for learning processes. In particular the cognitive and meta cognitive actions describe how learners control their thoughts, feelings and actions in order to achieve best learning results [3]. The use of efficient learning strategies is hence an essential prerequisite for academic success [4, 5]. According to van Lohuizen, the term 'learning strategy' is used for clusters of related learning activities that students have at their disposal in reaction to a specific learning goal [6]. And even though there are various classifications of learning strategies, three general scales emerged that describe cognitive, meta-cognitive, and resource-oriented strategies [7, 8]. Cognitive strategies serve to process the information collected e.g. in lectures and seminars. These cognitive strategies consist of organization, critical thinking, development and rehearsal of learning material. Meta-cognitive strategies help students to control and regulate their cognition and are subdivided into the subscales of planning (setting goals), regulating, and monitoring the learning processes [9]. Resource-oriented strategies differentiate intrinsic, and extrinsic resources and are divided into the intrinsic subscales of distractibility, effort regulation and time management while extrinsic subscales include managing the study environment, peer-learning, and use of additional literature.

Studying medicine is considered to be particularly demanding and learning-intensive. Students need to be proficient in defining their own learning goals, acquiring new knowledge and skills independently, and assessing the outcome of the learning process. Importantly, efficient learning is not only key for passing the upcoming examinations and successful completion of the medical degree, but provides the basis for lifelong professional advancement and keeping up with current scientific knowledge [1, 10]. Successful and sustained learning is therefore mandatory for the long-term provision of high-quality patient care [11–13].

Various studies have previously addressed the learning strategies medical students use to cope with university requirements [14, 15]. In the project described in this manuscript, we explored the learning strategies used by medical students at our university during their preclinical years. We investigated whether these learning strategies were constant or changed over time and whether they correlated with academic success. To that extend, we performed an exploratory study and recruited medical students during their first preclinical year. We assessed their individual learning strategies just before their first examination period at three months into medical school and repeated this assessment one year later. By analyzing the results of examinations written shortly after these assessments, we were able to indirectly correlate the various learning strategies with academic success.

# Methods

## Participant recruitment

During a compulsory course in their first preclinical year, a total of 224 students at the Rostock University Medical Center was informed about the study and invited to participate. Participation required written consent to the monitoring of learning approaches and examination scores as well as the participation in a self-assessment that contained demographic data. As an incentive to participate, we assured the students access to their results and evaluations at the end of the study. The study has received approval from the ethics committee of the medical faculty of the University of Rostock, it is registered under A 2018–0005 and was performed in

accordance with the Declaration of Helsinki. Written informed consent was obtained from all participants.

## Learning strategies

The learning strategies were assessed using the LIST-Questionnaire (Lernstrategien im Studium) which is a German version based on the Anglo-American Motivated Strategies of Learning questionnaire (MSLQ) [16, 17]. The LIST contains 96 items that score on an endpoint named Likert-scale ranging from 1 ('not at all true of me´) to 5 ('very true of me´) and classifies various learning activities into the following four domains–cognitive strategy use, meta-cognitive strategy use, intrinsic and extrinsic resource-oriented strategy uses. The LIST was compiled as an online questionnaire using EvaSys and sent to the students by email in their first (T1) and second (T2) preclinical year.

## Learning outcome

The medical school at Rostock University adheres to a traditional curriculum, teaching each subject individually and segregating the first two preclinical form the four clinical years. During the preclinical years, the students attend lectures and seminars, possibly take smaller examinations in the course of the term and through passing these, gain access to the final examinations at the end of the term. Smaller subjects like Biology stretch over the first term, require attending a lecture as well as practical courses and conclude with a written examination. More comprehensive subjects like physiology cover the third and fourth term and require attending lectures as well as seminars during the third term. The written examinations at the end of the third term is prerequisite for participating in a practical course in the fourth term which is flanked by oral examinations. Once all credits associated with the preclinical years are obtained, students can register for the first state examination which in turn needs to be passed in order to move on to the clinical years. However, students can opt out of any preclinical examination–at the cost of having to extend the preclinical phase.

 To monitor the learning success, we here used the results of the biology examination at T1 and the physiology examination at T2, respectively. The biology examination consisted of 40 multiple choice questions and the physiology examination of 75 multiple choice plus 5 open-ended questions. The mean scores achieved by a cluster of students using the same learning strategies were then compared.

 Simultaneous to the LIST, the participants received a second questionnaire asking them how they rated their own learning outcomes, how much time they spent studying and to predict their examination results. These self-evaluations were later compared to the real scores achieved. Moreover, the students were asked about their age, sex and final school grades.

## Data analysis

Fisher's exact tests were performed to compare the ratio of female to male students between the study cohort and the rest of the academic year. For each participant, the individual LIST-scores describing one of the thirteen subscales were totaled and entered into hierarchical cluster analyses (Ward method). Exploratory data analyses of the individual clusters resulted in the mean LIST-scores for each of the thirteen subscales. We conducted Kolmogorov-Smirnov tests to analyze the Gaussian distribution of the data. Comparisons between the mean LIST-scores and the corresponding demographic data were performed via one-way analysis of variance (ANOVA) (Scheffé-procedure) and Kruskal-Wallis tests, respectively. Comparisons of examination results between the various learning strategy combinations were computed via Mann-Whitney (biology) and unpaired t-tests (physiology examination), respectively. P-

values < 0.05 were considered statistically significant. Reliability was calculated via Cronbach´s alpha. All analyses were carried out using IBM SPSS Statistics, version 25.

## Results

### Study cohort

Fig 1 summarizes the numbers of participants at the various stages of the study as well as a time line. Out of a total of 224 first year medical students 157 students initially declared their consent, 123 of these filled out the LIST at the first time point (T1) and sat the biology examination. 100 of these students completed the self-assessment at T1. Of the 123 LIST participants at T1, 109 students completed the LIST at the second time point (T2), and 107 of these sat the physiology examination, so that the corresponding results were available. 103 of these study participants completed the self-assessment at T2. Reasons for being excluded from further analyses were the lack of or incompletely filled in LIST-questionnaires. The ratio of male/female students in the study cohort was 29/78 and was 46/71 for the rest of the academic year. The difference between male/female proportions in both cohorts was not statistically significant (p-value = 0.065).

### Students in their first year at medical school segregated into four distinct patterns of learning strategy combinations

The LIST comprises 96 items that classify various learning activities into four scales. The cognitive scale differentiates 4 subscales, organization, critical thinking, elaboration, and rehearsal of learning material. The meta-cognitive scale differentiates 3 subscales, planning, regulating, and monitoring the learning process. The resource-oriented scales differentiate 3 intrinsic and extrinsic strategies each, distractibility, effort regulation, and time management on the one hand and managing of the study environment, peer-learning and use of additional literature on the other. The reliability for each of the scales was calculated as Cronbach's alpha and was 0.85 for cognitive learning strategies, 0.68 for meta-cognitive learning strategies, 0.76 for intrinsic, and 0.79 for extrinsic resource-oriented strategies.

The individual LIST-scores describing one of the thirteen subscales were totaled and entered into hierarchical cluster analyses. These analyses were performed for T1 and T2 and at

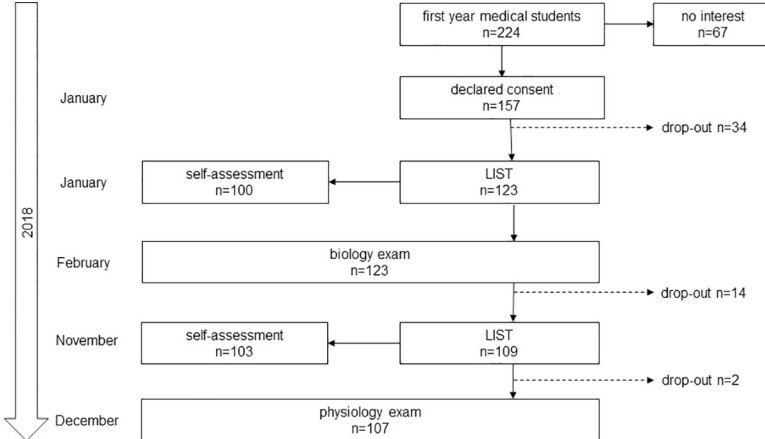

**Fig 1. The flow chart summarizes the numbers of study participants at entry, the various time points of data collection, and the drop-outs.**

both time points yielded four identical clusters. These clusters identified distinct patterns of learning strategy combinations among medical students and defined 4 types of learners.

The *relaxed learners* impressed with their significantly elevated LIST-scores regarding distractibility and were followed by sociable learners (Fig 2A). As for all the other subscales, except for critical thinking, the relaxed learners achieved the lowest scores of all types of learners. Moreover, their self-assessed learning outcomes concerning the upcoming examination revealed a rather relaxed attitude that prompted their name. At T1, 17 students belonged to the relaxed type of learners. The *diligent learners* impressed with the highest scores in all cognitive, meta-cognitive, and intrinsic resource-oriented subscales and only marginally decelerated when it came to managing the study environment. They invested the highest efforts into critical thinking (Fig 2B), organizing, and elaborating the subjects at hand and spent the most time planning, regulating, and monitoring their learning activities. Because they not only invested

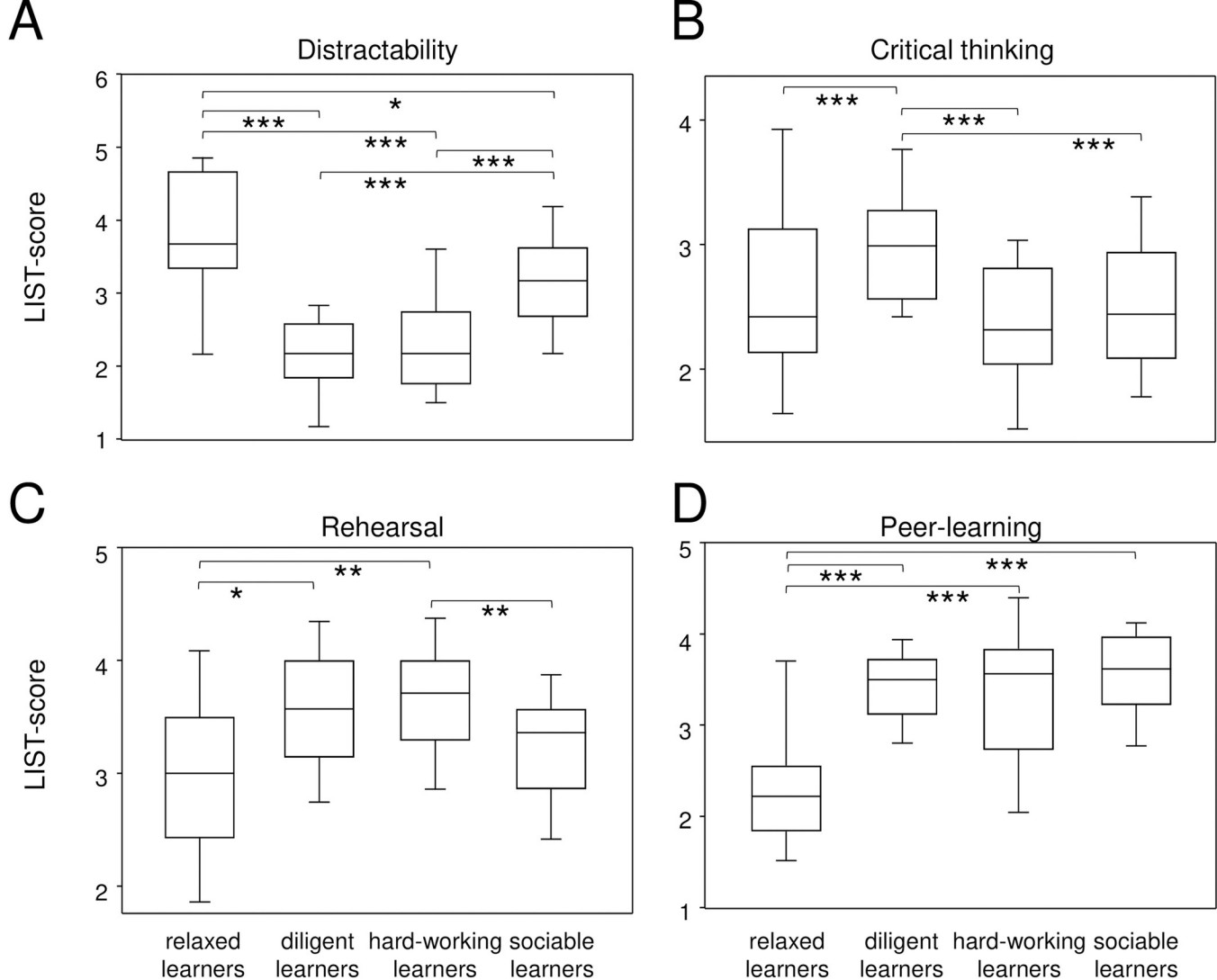

**Fig 2. The predominant subscales define four distinct patterns of learning strategy combinations among first year medical students.** Box-plots show LIST scores of the most characteristic learning activities defining the four types of learners at T1. Statistical significance resulting from ANOVA (Scheffé-test with $\alpha$ = 0.05) are indicated by asterisks. $^{*}$p<0.05; $^{**}$p<0.01; $^{***}$p<0.001.

**Table 1. Learning strategies among medical students segregate into four distinct patterns.**

|  | relaxed (n = 17) | diligent (n = 25) | hard-working (n = 33) | sociable (n = 48) |  |
|---|---|---|---|---|---|
|  | Mean [±SD] | Mean [±SD] | Mean [±SD] | Mean [±SD] | P-Value# |
| Organization | 3.0 [0.63] | 4.1 [0.47] | 3.8 [0.68] | 3.6 [±0.50] | $8.9 \times 10^{-8}$ |
| Elaboration | 3.2 [0.52] | 4.0 [0.42] | 3.2 [0.39] | 3.5 [±0.51] | $3.8 \times 10^{-9}$ |
| Planning the learning process | 2.8 [0.61] | 3.7 [0.34] | 3.3 [0.47] | 3.1 [±0.53] | $5.0 \times 10^{-8}$ |
| Regulating the learning process | 2.9 [0.53] | 4.1 [0.45] | 3.3 [0.50] | 3.5 [±0.60] | $2.6 \times 10^{-9}$ |
| Monitoring the learning process | 4.0 [0.49] | 4.4 [0.41] | 4.4 [0.39] | 4.1 [±0.50] | $4.6 \times 10^{-3}$ |
| Effort regulation | 3.2 [0.45] | 4.2 [0.41] | 4.1 [0.36] | 3.6 [±0.49] | $2.4 \times 10^{-13}$ |
| Time management | 1.9 [0.57] | 3.6 [0.48] | 3.2 [0.84] | 2.5 [±0.51] | $1.8 \times 10^{-16}$ |
| Study environment | 3.3 [0.57] | 4.3 [0.43] | 4.4 [0.38] | 3.8 [±0.53] | $4.6 \times 10^{-13}$ |
| Use of additional literature | 3.7 [0.76] | 4.2 [0.66] | 4.0 [0.58] | 4.0 [±0.71] | 0.1 |

#according to ANOVA (Scheffé-test with $\alpha = 0.05$); standard deviation (SD).

so much effort but also used their resources diligently, we named them the diligent learners. At T1, 25 students belonged to the diligent type of learners. The *hard-working learners* also scored high on learning strategies that require cognitive and meta-cognitive skills. However, they focused on repetitive activities like rehearsal (Fig 2C) and placed high emphasis on an efficient study environment yet scored the lowest on critical thinking. We, therefore, named them the hard-working learners and at T1, 33 students were the hard-working type. And finally, there were 48 *sociable learners* at T1. They preferred to learn in the company of others and scored the highest with the extrinsic resource-oriented strategy of peer-learning (Fig 2D). We called these the sociable learners.

Table 1 summarizes the mean individual scores achieved for the four different patterns of learning strategy combinations and the remaining subscales not presented in Fig 2. The p-values resulting from ANOVA confirm four distinct patterns of learning strategy combinations. The statistics regarding self-assessment and demographics of the four different patterns of learning strategy combinations revealed that the diligent and the hard-working learners were the youngest and spent the most time studying (Table 2). The difference in school leaving grades showed a tendency but did not reach statistical significance.

## Learning patterns were flexible but defined preferences

When repeating the hierarchical cluster analysis of LIST-scores at T2, the same four patterns were reproduced. However, only half of the students stayed true to their previous learning habits while 53/107 study participants changed their learning strategies in the course of the pre-clinical training (Fig 3). At close inspection though, the changes did not occur randomly. Instead, certain preferences emerged that appeared stable. E.g., while most of the relaxed learners (10/12) remained relaxed, the two who did change turned into sociable learners. Likewise,

**Table 2. Demographics of learning patterns.**

|  | relaxed (n = 17) | diligent (n = 25) | hard-working (n = 33) | sociable (n = 48) |  |
|---|---|---|---|---|---|
| female/male (n) | 10/7 | 19/6 | 22/11 | 41/7 |  |
|  | Mean [±SD] | Mean [±SD] | Mean [±SD] | Mean [±SD] | P-Value# |
| age | 23.2 [±3.15] | 19.3 [±1.98] | 20.4 [±2.57] | 20.7 [±3.20] | $9.5 \times 10^{-13}$ |
| final school grade | 1.8 [±0.50] | 1.3 [±0.33] | 1.4 [±0.40] | 1.5 [±0.51] | $5.7 \times 10^{-2}$ |
| self-study (hrs/week) | 13.5 [±5.5] | 24.0 [±4.0] | 21.0 [±6.0] | 16.5 [±5.5] | $7.4 \times 10^{-4}$ |

#according to ANOVA (Scheffé-test with $\alpha = 0.05$). Note, that higher grades in Germany represent lower academic performance.

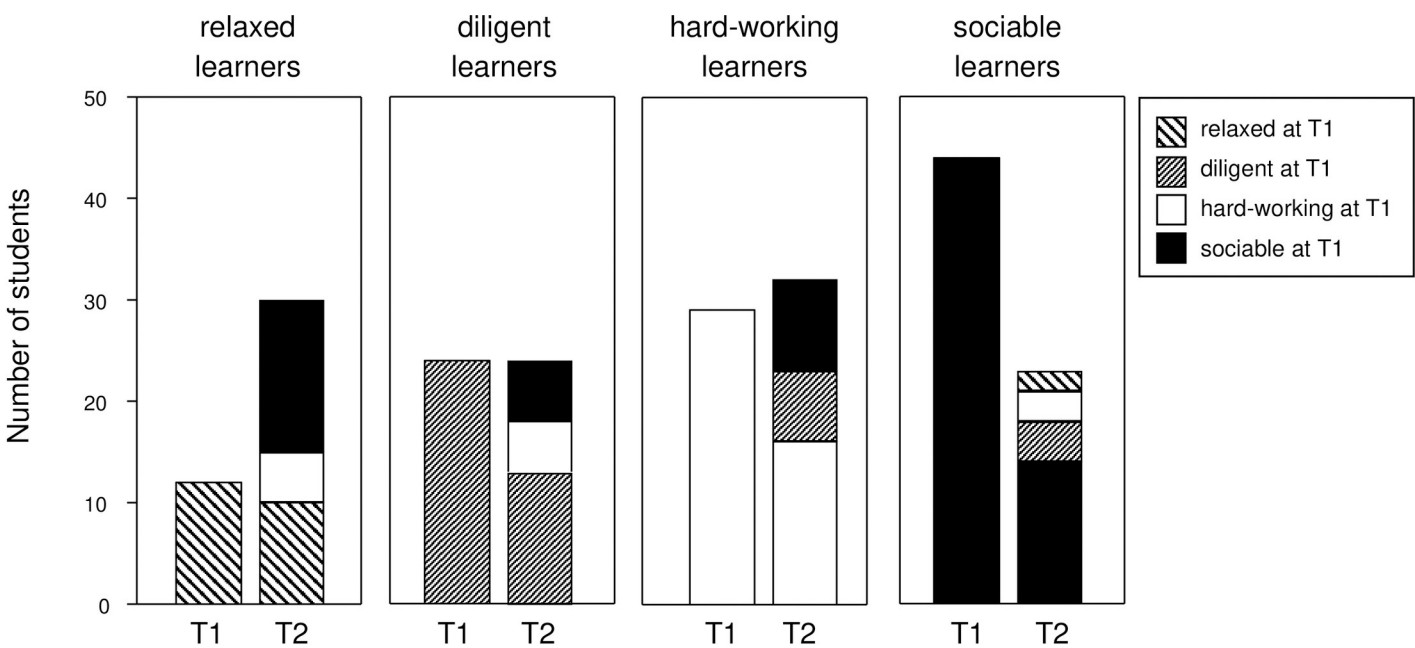

**Fig 3. Learning patterns were flexible but defined preferences.** Bar graphs show changes of learning patterns in the course of the first 18 months at medical school. The graphic designs define the learning patterns at T1 and indicate changes towards different patterns at T2. Only those students were included for whom T2 data were fully available.

most of the sociable ones remained sociable (14/44) or ended up relaxed (n = 15). Only 6 of the originally relaxed learners turned out diligent. In summary, the relaxed learners recorded the strongest growth, while the sociable ones were reduced by half.

On the other hand, the majority of the diligent learners remained diligent (13/24) and of those who changed, the majority (n = 7) became hard-working. None turned into a relaxed learner. Among the hard-working learners at T1 (n = 29), 16 remained hard-working and 5 became diligent, while 3 turned into sociable learners and 5 became relaxed. The preferences emerging were thus either hard-working and/or diligent on the one hand or social and/or relaxed on the other.

## Academic success segregated with the *hard-working* and *diligent* learners

In order to analyze whether certain learning patterns were academically more successful and led to better examination results, we compared the various patterns for the scores achieved in the biology and physiology examinations. There were no significant differences when the four learning patterns were compared to each other. However, when grouping according to learning preferences–the relaxed and the sociable learners versus the diligent and hard-working ones–the latter obtained significantly better scores and that was true for the biology as well as the physiology examinations (Fig 4). The combined medians for the scores obtained in the biology examination at T1 were 34 (interquartile range IQR = 6) and 35 (IQR = 5) for the sociable/relaxed and the diligent/hard-working learners, respectively. For the physiology examination they were 48 (IQR = 14) and 53.5 (IQR = 10).

## Discussion

We here described four distinct patterns of learning strategy combinations among medical students which were found in repeated assessments yet for the individual student were not

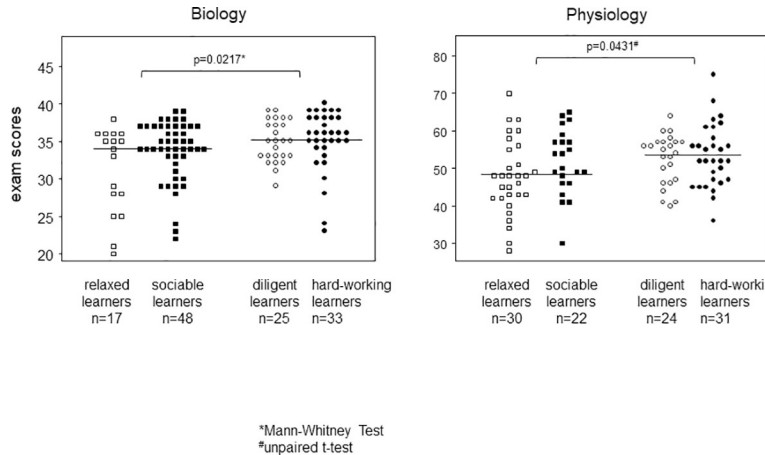

**Fig 4. Academic success segregated with the *diligent* and *hard-working* learners.** Dot-plots show the scores obtained for the relaxed/sociable and the hard-working/diligent learners in the biology examination at T1 and the physiology examination at T2. Statistical significance was calculated via Mann-Whitney (biology) and unpaired t-test (physiology examination) and is indicated by p-values.

necessarily stable. Students were first assessed six months into medical school and at this point in time, we consider it likely that the individual learning strategy combined an individual´s preference with the strategy that had proved itself successful in high school. A student´s learning approach may be regarded as a combination of several components, among them views about learning as well as metacognitive and processing activities, whereby the actualized learning strategy is assumed to depend on the specific learning situation [18, 19]. Indeed, at the second assessment conducted one year later, 49.5% of the students had changed their learning habits. Whether this change occurred on purpose because the students were afraid of failure or whether the change was unintentional we cannot conclude. However, the observation that changes in learning patterns did not occur at random but predominantly between the sociable and relaxed learners on the one hand and between the hard-working and diligent ones suggested certain preferences. We therefore investigated whether these preferences correlated with academic success. And indeed, the preference for a sociable or relaxed learning pattern turned out significantly less successful compared to the preference for a diligent or hard-working one. Our results therefore support the notion that the right choice of learning activities influences academic success [20].

It is not surprising that most of these early changes in learning patterns were made by the sociable learners, given the more time-consuming schedule that leaves little time for group meetings in the second preclinical year. And even though comprehensible, any decrease in sociability will hamper the fostering of important skills like team work and will therefore reduce motivation, which in turn stimulates discussion and critical thinking about the learning material [21, 22]. Unfortunately, the sociable learning strategy was frequently replaced with the relaxed one, indicating an individual preference for unfavorable learning strategies.

Indeed, the relaxed learners not only featured the most inefficient pattern of learning strategy combinations, they also spent the least time for self-study. A glance at their final school grades suggested that these students already in their earlier years had learning deficiencies. And even when abandoning their old learning preference, the relaxed ones remained true to an inefficient pattern and predominantly turned into sociable learners. Distressingly, the numbers of relaxed learners doubled between the first and the second survey, suggesting an increase of students who were bound to run into academic problems in the years to come.

The most significant differences between the four patterns of learning strategy combinations were caused by differences in resource-oriented learning strategies like distractibility, time management and effort. Here the diligent learners achieved particularly low LIST-scores for the former and particularly high scores for the latter two. They thus appeared well prepared for the requirements of medical school. Moreover, the diligent learners scored also highest in many other learning activities that allowed for a flexible realignment to various learning problems. The fact that they also achieved highest scores in cognitive and metacognitive learning strategies (except for repetition) indicated a preferred learning to understand as opposed to superficial memorizing. Indeed, previous publications showed that medical students who pursue a deep or strategic learning approach achieve highest academic success, whereas a surface approach correlated with poorer outcomes [4, 23, 24].

The hard-working learners relied on repetitive strategies, however they lacked the capacity to apply a variety of learning activities. On the long run, this is bound to turn out negative with respect to the numerous theoretical and practical requirements of the clinical years and the professional life thereafter. We favor the idea that the hard-working learners changed their patterns of learning strategy combination most flexibly because they realized at some point that their strategies did not suffice to keep up with the increasing learning demands and were in dire need of an alternative strategy.

Both, the relaxed and the sociable learners are likely to benefit from learning skill interventions [25]. Medical educators therefore need to impart sufficient metacognitive knowledge to adopt the hard-working if not the diligent strategy and to show students how to gain control over their learning processes in order to turn into medical doctors that are proficient in lifelong learning [1].

There are limitations to our study and these concern the relatively small sample size of 107 students and even smaller subgroups for the various patterns of learning strategy combinations. Moreover, our monocentric design and the short observational period within the framework of a traditional curriculum limit the explanatory power. It would be interesting to find out how the learning habits differ in a problem-based teaching environment and how they develop over longer time, in particular, during the clinical education. As we here compared learning patterns to the biology and physiology examinations only, we cannot extrapolate which learning strategy would be the best fit for other preclinical, let alone clinical subjects. Moreover, any questionnaire assessing learning strategies harbors the risk of socially desirable statements. However, as opposed to the available literature which assessed academic success via cumulative grade point averages and temporal distance, we here evaluated examinations written in temporal proximity to the surveys on learning strategies [4, 5, 14]. We therefore were not only able to follow individual developments but also to directly correlate changes in the learning strategies to academic achievements.

In summary, we here defined four distinct patterns of learning strategy combinations among early medical students that were defined by differences concerning resource-oriented learning strategies. Early habits of sociable learning were quickly abandoned however, not in favor of more successful patterns. It is therefore essential to develop interventions on learning skills that have a lasting impact on the pattern of learning strategy combinations.

## Acknowledgments

The authors are grateful to all study participants for their readiness to give us an insight into their learning behavior during the preclinical years. Moreover, we would like to express our thanks to the members of the student research group for their critical comments during data analysis.

## Author Contributions

**Conceptualization:** Annemarie Hogh, Brigitte Müller-Hilke.

**Data curation:** Annemarie Hogh, Brigitte Müller-Hilke.

**Formal analysis:** Annemarie Hogh, Brigitte Müller-Hilke.

**Investigation:** Annemarie Hogh, Brigitte Müller-Hilke.

**Methodology:** Annemarie Hogh, Brigitte Müller-Hilke.

**Project administration:** Brigitte Müller-Hilke.

**Resources:** Brigitte Müller-Hilke.

**Software:** Brigitte Müller-Hilke.

**Supervision:** Brigitte Müller-Hilke.

**Validation:** Annemarie Hogh, Brigitte Müller-Hilke.

**Visualization:** Annemarie Hogh, Brigitte Müller-Hilke.

**Writing – original draft:** Annemarie Hogh, Brigitte Müller-Hilke.

**Writing – review & editing:** Annemarie Hogh, Brigitte Müller-Hilke.

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
