## [Decision Letter · Decision Letter 0]

29 Oct 2020

PONE-D-20-27667

Learning strategy as predictor of academic success in medical school

PLOS ONE

Dear Dr. Müller-Hilke,

Thank you for submitting your manuscript to PLOS ONE. After careful consideration, we feel that it has merit but does not fully meet PLOS ONE’s publication criteria as it currently stands. Therefore, we invite you to submit a revised version of the manuscript that addresses the points raised during the review process.

The reviewers had concerns about the title, the grading methods, the review of the literature and the context as well as the limitations of the work.

We look forward to receiving your revised manuscript.

Kind regards,

Mohammed Saqr, Ph.D

Academic Editor

PLOS ONE

Journal Requirements:

2.Thank you for including your ethics statement:

"The study was approved by the local ethics committee and is registered under A 2018-0005."

Reviewers' comments:

Reviewer's Responses to Questions

**Comments to the Author**

1. Is the manuscript technically sound, and do the data support the conclusions?

Reviewer #1: Yes

Reviewer #2: Partly

2. Has the statistical analysis been performed appropriately and rigorously? 

Reviewer #1: Yes

Reviewer #2: Yes

3. Have the authors made all data underlying the findings in their manuscript fully available?

Reviewer #1: Yes

Reviewer #2: Yes

4. Is the manuscript presented in an intelligible fashion and written in standard English?

Reviewer #1: Yes

Reviewer #2: Yes

5. Review Comments to the Author

Reviewer #1: The manuscript is well written and presents a very interesting topic. The authors present the subject matter in a clear and concise way that captures the reader's interest.

I am missing a review of related work on medical education or similar studies in other fields. In addition, I think the study is lacking in theoretical foundation. For instance, the authors use the LIST questionnaire, which is based on Piltrich's MSLQ instrument for self-regulation, but they do not mention the concept of self-regulation at all in their manuscript as the basis of the learning process (cognitive and meta cognitive actions). I think including such theoretical foundation for students' actions would greatly help position the article within existing literature.

Some statements in the result analyisis are a bit "bold". For instance, in line 146, "For the comparison of both cohorts resulted in a p-value of 0.065, confirming comparable ratios". A p-value of 0.065 fails to reject the null hypotheses, but does not mean that the opposite is confirmed. It should say something like "The difference between male/female proportions in both cohorts was not statistically significant (p-value = .065)". Also, in Table 1, for instance, for such low p-values, it is usually specified as p < .001 instead of including so many decimal figures.

Lastly, I think it would be useful to include some sort of timeline of the study, including the interventions used at each point in time.

Reviewer #2: This study looks at different study strategies adopted by preclinical medical students at a German university and investigates whether these choices impact the academic performance for two basic science topics, biology and physiology. The authors also analyze whether students change their strategy from the first to the second year of medical school. The authors mention that they see learning success (examination scores) differences between groups of students with different learning strategies, but none of these differences is statistically significant. That may be due to the small sample size. The second question about students changing their learning strategy would also be of interest, but again it is hampered by the small sample size.

These drawbacks of the study make it very preliminary and we can’t draw any useful conclusions at this time. I would advise the authors to collect more data and if no significant learning success differences between learning strategies become apparent, to concentrate on the second question. It would be of interest to evaluate why students change their learning strategy and whether that makes a positive, negative or no difference for their academic success of these students.

Nice short title that describes that topic and the goal of the study. I would advise to reword the title and to avoid the word “predictor”. What is the authors look at is a correlation (and they do not find any).

The introduction is well organized, concise and develops the problems addressed in this project.

The English of the manuscript, specifically the abstract, could be improved. Some of the word choices, although not wrong, are suboptimal. See a few examples below.

For example: “We here explored the learning strategies applied during the preclinical years, whether they are constant traits or subject to change and how they impact on the academic success.”

Better: “In the project described in this manuscript we explored the learning strategies used by medical students at our university during their preclinical years. We investigated whether these learning strategies were constant or changed over time and whether they correlated with academic success.”

The abstract misrepresents the finding of the study. As academic differences between different study strategies are not statistically significant, we can’t draw any conclusion as to which strategy/ies is/are best. The conclusion section is not based on the data, but are simple truisms that are independent of the results presented in the described work.

The Methods section is missing a lot of important information.

I would appreciate more background about the structure of the German preclinical curriculum as it is used at the University of Rostock. Clearly describe the situation, like the size of the entire class (224) and its composition? Were the study participants a representative sample of the class? Which students were eliminated from the study and for what reasons. How and when were the surveys offered? There is only a very superficial description of how the learning outcome was measured. What is the role of the self-assessment? It appears that only results of a small biology/physiology examinations were used, leaving open that other learning strategies might be a better fit for other topics (anatomy, biochemistry, pharmacology etc.). It would improve the analysis if additional or more general examination scores would be used for the correlation with the different learning strategies used by different students.

The statistical analysis appears to be appropriate.

Another significant problem of the study is that the academic success was measured using different examination topics (as this is a longitudinal study, there is no way to change this aspect), biology for year 1 and physiology for year 2. At least this needs to be critically discussed.

Also, it is never clearly stated that both topics show no statistically significant differences for the different learning strategies. That may be a direct result of the small sample size. You can’t discuss the differences in the result section if they are not statistically significant.

The most interesting result is that many students keep their initial study strategy, although some change. Again, the small sample size does not allow a more detailed analysis why these changes occur and how they impacted individual student’s performance.

Good that the authors included a paragraph about the limitations of their study. I would add a title “Limitation of the study” and organize this paragraph as a subsection of the discussion.

Limitations are that this analysis looks at preclinical learning, not on the students clinical abilities. There is nothing wrong with that, however, this limitation needs to be stated. As the examinations only cover biology/physiology, do NOT use the general term “academic success”, but make clear that it is only academic success in these subjects. Without further information and analysis, academic success can’t be assumed for other preclinical and clinical subjects.

Smaller issues:

Define acronyms at their first appearance and independently in the abstract. E.g., LIST is never defined.

“Medical school” should be “medical school”.

“Examinations”, not “exams”.

“Likert”, not “LIKERT”. Likert is a name (of the social psychologist Rensis Likert), not an acronym.

Not “MC-questions”, but the correct acronym is MCQ for Multiple Choice Question.

“…and sat the biology exam.” That is not a complete sentence.

6. PLOS authors have the option to publish the peer review history of their article (what does this mean?). If published, this will include your full peer review and any attached files.

Reviewer #1: No

Reviewer #2: No

---

## [Author Response · Author response to Decision Letter 0]

8 Dec 2020

PONE-D-20-27667

Learning strategy as predictor of academic success in medical school

PLOS ONE

Dear Reviewers,

We highly appreciate your comments - as they did point out (potential) misunderstandings and weaknesses. In fact, they prompted us to look at our data again with fresh eyes – and to revise our manuscript accordingly. We believe it improved significantly.

Below, please find our point by point reply.

Reviewer #1: The manuscript is well written and presents a very interesting topic. The authors present the subject matter in a clear and concise way that captures the reader's interest.

I am missing a review of related work on medical education or similar studies in other fields. In addition, I think the study is lacking in theoretical foundation. For instance, the authors use the LIST questionnaire, which is based on Piltrich's MSLQ instrument for self-regulation, but they do not mention the concept of self-regulation at all in their manuscript as the basis of the learning process (cognitive and meta cognitive actions). I think including such theoretical foundation for students' actions would greatly help position the article within existing literature.

completely agreed – we introduced another sentence plus reference, please see LL59-61

Some statements in the result analyisis are a bit "bold". For instance, in line 146, "For the comparison of both cohorts resulted in a p-value of 0.065, confirming comparable ratios". A p-value of 0.065 fails to reject the null hypotheses, but does not mean that the opposite is confirmed. It should say something like "The difference between male/female proportions in both cohorts was not statistically significant (p-value = .065)". Also, in Table 1, for instance, for such low p-values, it is usually specified as p < .001 instead of including so many decimal figures.

Agreed – and done! Please see LL 105/106

Lastly, I think it would be useful to include some sort of timeline of the study, including the interventions used at each point in time.

Also agreed – and done. Please see Figure 1.

Reviewer #2: This study looks at different study strategies adopted by preclinical medical students at a German university and investigates whether these choices impact the academic performance for two basic science topics, biology and physiology. The authors also analyze whether students change their strategy from the first to the second year of medical school. The authors mention that they see learning success (examination scores) differences between groups of students with different learning strategies, but none of these differences is statistically significant. That may be due to the small sample size. The second question about students changing their learning strategy would also be of interest, but again it is hampered by the small sample size.

These drawbacks of the study make it very preliminary and we can’t draw any useful conclusions at this time. I would advise the authors to collect more data and if no significant learning success differences between learning strategies become apparent, to concentrate on the second question. It would be of interest to evaluate why students change their learning strategy and whether that makes a positive, negative or no difference for their academic success of these students.

We considered your comments for quite a while – and that made us look at our data with fresh eyes. The preferences of our students – if they change their learning pattern, they do so mainly between the social and relaxed or between the diligent and hard-working patterns – made us compare the examination results accordingly. And that did indeed reveal academically more successful (diligent and hard-working) and less successful (social and relaxed) patterns. We revised our manuscript accordingly. 

Nice short title that describes that topic and the goal of the study. I would advise to reword the title and to avoid the word “predictor”. What is the authors look at is a correlation (and they do not find any).

Agreed and done!

The introduction is well organized, concise and develops the problems addressed in this project.

The English of the manuscript, specifically the abstract, could be improved. Some of the word choices, although not wrong, are suboptimal. See a few examples below.

For example: “We here explored the learning strategies applied during the preclinical years, whether they are constant traits or subject to change and how they impact on the academic success.”

Better: “In the project described in this manuscript we explored the learning strategies used by medical students at our university during their preclinical years. We investigated whether these learning strategies were constant or changed over time and whether they correlated with academic success.”

Agreed and done! Please, see LL83-86

The abstract misrepresents the finding of the study. As academic differences between different study strategies are not statistically significant, we can’t draw any conclusion as to which strategy/ies is/are best. The conclusion section is not based on the data, but are simple truisms that are independent of the results presented in the described work.

Revised (according to our comment above).

The Methods section is missing a lot of important information.

I would appreciate more background about the structure of the German preclinical curriculum as it is used at the University of Rostock. Clearly describe the situation, like the size of the entire class (224) and its composition? 

We revised the “study cohort” paragraph in the results section (LL96-106) as well as the “learning outcome” paragraph in the methods section (LL 309-321) in order to provide the missing information.

Were the study participants a representative sample of the class? 

In terms of age and sex: yes. Other than that: we assume yes - but would not know.

Which students were eliminated from the study and for what reasons. 

We excluded those students, whose data were incomplete. E.g. they may have declared consent but then did not even fill in the first LIST (n=34), they may have had complete data sets for T1 (the first LIST and biology examination scores) but did not respond at T2 (n=14) or filled in the second LIST but then did not write the physiology examination (n=2). We rephrased in the Results section, please see LL 102/103.

How and when were the surveys offered? There is only a very superficial description of how the learning outcome was measured. What is the role of the self-assessment? It appears that only results of a small biology/physiology examinations were used, leaving open that other learning strategies might be a better fit for other topics (anatomy, biochemistry, pharmacology etc.). It would improve the analysis if additional or more general examination scores would be used for the correlation with the different learning strategies used by different students.

The statistical analysis appears to be appropriate.

There was only one examination each in temporal proximity to the LIST surveys. That was for one a chemistry examination close to the biology one. However to our experience, the results of the chemistry exam are very much dependent on whether and for how long students had had chemistry in school. We therefore refrained from including the chemistry examination. For the other, there is the biochemistry examination shortly after the physiology examination. However, the physiology examination among students is considered the more challenging one – leading to a high percentage of failures in biochemistry, because students concentrate their learning activities on physiology. At the time we therefore decided to evaluate the physiology results. Looking back it would have been wise to include all results. As it is, we cannot simply include chemistry and biochemistry results, because the written consent explicitly asked for permission to evaluate biology and physiology.

At the end of the “learning outcome” paragraph in the methods section we elaborate on the self-evaluations. (LL 327-330)

Another significant problem of the study is that the academic success was measured using different examination topics (as this is a longitudinal study, there is no way to change this aspect), biology for year 1 and physiology for year 2. At least this needs to be critically discussed.

We address this in our “limitations of the study” section, please see LL 265-278

Also, it is never clearly stated that both topics show no statistically significant differences for the different learning strategies. That may be a direct result of the small sample size. You can’t discuss the differences in the result section if they are not statistically significant.

The most interesting result is that many students keep their initial study strategy, although some change. Again, the small sample size does not allow a more detailed analysis why these changes occur and how they impacted individual student’s performance.

as pointed out above, we revised our manuscript and believe, that this particular comment is now obsolete

Good that the authors included a paragraph about the limitations of their study. I would add a title “Limitation of the study” and organize this paragraph as a subsection of the discussion.

Limitations are that this analysis looks at preclinical learning, not on the students clinical abilities. There is nothing wrong with that, however, this limitation needs to be stated. As the examinations only cover biology/physiology, do NOT use the general term “academic success”, but make clear that it is only academic success in these subjects. Without further information and analysis, academic success can’t be assumed for other preclinical and clinical subjects.

We have extended this paragraph (LL276-279) in order to consider your comments – but abstained from an extra title. We do not have any subtitles at all in our discussion – and considered it not appropriate to put this extra weight on our limitations.

Smaller issues:

Define acronyms at their first appearance and independently in the abstract. E.g., LIST is never defined.

“Medical school” should be “medical school”.

“Examinations”, not “exams”.

“Likert”, not “LIKERT”. Likert is a name (of the social psychologist Rensis Likert), not an acronym.

Not “MC-questions”, but the correct acronym is MCQ for Multiple Choice Question.

All agreed and done!

While revising your submission, please upload your figure files to the Preflight Analysis and Conversion Engine (PACE) digital diagnostic tool, 

DONE

---

## [Decision Letter · Decision Letter 1]

29 Dec 2020

PONE-D-20-27667R1

Learning strategies and their correlation with academic success in biology and physiology examinations during the preclinical years of medical school

PLOS ONE

Dear Dr. Müller-Hilke,

Thank you for submitting your manuscript to PLOS ONE. After careful consideration, we feel that it has merit but does not fully meet PLOS ONE’s publication criteria as it currently stands. Therefore, we invite you to submit a revised version of the manuscript that addresses the points raised during the review process.

As you can see in the reviewers comments, there are few minor comments that need to be addressed. You may also  need to consult PlOS one for styles guidelines before submitted this version.

We look forward to receiving your revised manuscript.

Kind regards,

Mohammed Saqr, Ph.D

Academic Editor

PLOS ONE

Reviewers' comments:

Reviewer's Responses to Questions

**Comments to the Author**

1. If the authors have adequately addressed your comments raised in a previous round of review and you feel that this manuscript is now acceptable for publication, you may indicate that here to bypass the “Comments to the Author” section, enter your conflict of interest statement in the “Confidential to Editor” section, and submit your "Accept" recommendation.

Reviewer #1: All comments have been addressed

Reviewer #2: (No Response)

2. Is the manuscript technically sound, and do the data support the conclusions?

Reviewer #1: Yes

Reviewer #2: Yes

3. Has the statistical analysis been performed appropriately and rigorously? 

Reviewer #1: Yes

Reviewer #2: Yes

4. Have the authors made all data underlying the findings in their manuscript fully available?

Reviewer #1: Yes

Reviewer #2: No

5. Is the manuscript presented in an intelligible fashion and written in standard English?

Reviewer #1: Yes

Reviewer #2: Yes

6. Review Comments to the Author

Reviewer #1: The authors have addressed all my previous concerns. The only remaining issue is the order of the sections. The Methods section should go before results, and not at the end of the manuscript.

Reviewer #2: I am very pleased with the corrections, clarifications and additions the authors have included in their revised manuscript. Their conclusions and the presentation are much clearer now. I think they understand that the small sample size and the limited number of topics reduce the impact of their study.

I have one remaining request, the real p-values need to be presented in Tables 1 and 2 and also for Figure 2. Just giving p-value ranges by number of stars is not sufficient.

7. PLOS authors have the option to publish the peer review history of their article (what does this mean?). If published, this will include your full peer review and any attached files.

Reviewer #1: No

Reviewer #2: No

---

## [Author Response · Author response to Decision Letter 1]

7 Jan 2021

Dear Reviewers,

A Happy New Year to you – and thank you for your friendly comments. Please, find below our responses:

Reviewer #1: The authors have addressed all my previous concerns. The only remaining issue is the order of the sections. The Methods section should go before results, and not at the end of the manuscript.

Done! 

Reviewer #2: I am very pleased with the corrections, clarifications and additions the authors have included in their revised manuscript. Their conclusions and the presentation are much clearer now. I think they understand that the small sample size and the limited number of topics reduce the impact of their study.

I have one remaining request, the real p-values need to be presented in Tables 1 and 2 and also for Figure 2. Just giving p-value ranges by number of stars is not sufficient.

Exact p-values have been reintroduced. Replacements by asterisks were introduced into the former version in order to accommodate a comment by Reviewer #1. We therefore feel a little bit between rock and had place…..

Thanks to both of you for your time and effort!

---

## [Editor Report · Decision Letter 2]

11 Jan 2021

Learning strategies and their correlation with academic success in biology and physiology examinations during the preclinical years of medical school

PONE-D-20-27667R2

Dear Dr. Müller-Hilke,

We’re pleased to inform you that your manuscript has been judged scientifically suitable for publication and will be formally accepted for publication once it meets all outstanding technical requirements.

Kind regards,

Mohammed Saqr, Ph.D

Academic Editor

PLOS ONE
---

## [Editor Report · Acceptance letter]

13 Jan 2021

PONE-D-20-27667R2 

Learning strategies and their correlation with academic success in biology and physiology examinations during the preclinical years of medical school 

Dear Dr. Müller-Hilke:

I'm pleased to inform you that your manuscript has been deemed suitable for publication in PLOS ONE. Congratulations! Your manuscript is now with our production department. 

Kind regards, 

on behalf of

Dr. Mohammed Saqr 

Academic Editor

PLOS ONE